# Development of a Targeted NGS Assay for the Detection of Respiratory Pathogens including SARS-CoV-2 in Felines

**DOI:** 10.3390/pathogens13040335

**Published:** 2024-04-17

**Authors:** Jobin J. Kattoor, Mothomang Mlalazi-Oyinloye, Sarah M. Nemser, Rebecca P. Wilkes

**Affiliations:** 1Animal Disease Diagnostic Laboratory, Department of Comparative Pathobiology, College of Veterinary Medicine, Purdue University, West Lafayette, IN 47907, USA; jkattoor@purdue.edu; 2Center for Veterinary Medicine, Vet-LIRN, Food and Drug Administration, Laurel, MD 20708, USA; m.mlalazi@magbiogenomics.com (M.M.-O.); sarah.nemser@fda.hhs.gov (S.M.N.)

**Keywords:** feline respiratory disease panel, SARS-CoV-2, targeted next-generation sequencing, ion torrent sequencing

## Abstract

Acute respiratory diseases in felines can be attributed to a diverse range of pathogens. The recent emergence of novel viruses, particularly SARS-CoV-2 and its variants, has also been associated with respiratory ailments in cats and other pets, underscoring the need for a highly sensitive diagnostic assay capable of concurrently detecting multiple respiratory pathogens. In this study, we developed a targeted next generation sequencing panel using Ion Torrent Ampliseq technology to detect multiple respiratory pathogens, including recent SARS-CoV-2 variants and Feline herpesvirus-1, Feline calicivirus, *Bordetella bronchiseptica*, *Mycoplasmopsis* (previously *Mycoplasma*) *felis*, and *Chlamydia felis*. A PCR amplification-based library preparation, employing primers designed for pathogen target regions, was synthesized and divided into two pools, followed by sequencing and assembly to a repertoire of target pathogen genomes. Analytical sensitivity was assessed based on Ct values from real-time PCR for the corresponding pathogens, indicating an equivalent detection limit. Most of the pathogens under study were positively identified to a limit of approximately Ct 36, whereas for Feline herpesvirus-1 and SARS-CoV-2, positive reads were observed in samples with a Ct of 37. Based on a limited number of samples, the diagnostic sensitivity values for the SARS-CoV-2, Feline herpesvirus-1, and *M. felis* samples were 100% with no false negative results. The diagnostic specificity of SARS-CoV-2, Feline herpesvirus-1, Feline calicivirus, and *C. felis* were 100%. Importantly, none of the target primers exhibited non-specific amplification, ensuring the absence of false positive results for other pathogens within the study. Additionally, the assay’s specificity was validated by cross-referencing the raw sequencing data with established databases like BLAST, affirming the high specificity of the targeted Next-Generation Sequencing (tNGS) assay. Variations in the sequencing reads of different pathogens were observed when subjected to diverse extraction methods. Rigorous assessment of the assay’s reliability involved reproducibility across testing personnel and repeated runs. The developed assay’s clinical applicability was tested using samples submitted to the diagnostic laboratory from cat shelters and suspected cases. The developed targeted next-generation sequencing methodology empowers the detection of multiple respiratory pathogens manifesting similar clinical symptoms while offering confirmation of results through genome sequencing.

## 1. Introduction

Respiratory infections are an important cause of morbidity and mortality, especially in kittens and sheltered cats of all ages. Crowded settings in animal shelters can lead to more serious disease conditions involving organisms such as Feline herpesvirus (FeHV-1), which otherwise are common inhabitants of the feline respiratory tract [1]. Co-infections with multiple respiratory pathogens are common in feline upper respiratory disease (URD), and can lead to severe illness [1]. Advances in sequencing technologies have unveiled numerous novel respiratory viruses in cats; the major pathogens causing feline URD are FeHV-1, Feline calicivirus (FCV), *Bordetella bronchiseptica*, *Chlamydia felis*, and *Mycoplasmopsis (Mycoplasma) felis* [2]. Other less common organisms are influenza and SARS-CoV-2 [2,3]. Most respiratory diseases manifest as flu-like symptoms, including sneezing, labored breathing, reduced appetite, fever, and malaise, which cannot be attributed to a single specific pathogen, and hence treatment is difficult without the actual identification of the pathogen. Currently, real-time PCR (qPCR) assays are considered the gold standard for the identification of upper respiratory tract pathogens in cats [4]. Although qPCR is the gold standard, the assay should be performed for each individual organism and hence is time-consuming, and sometimes the laboratory may not be equipped with the diagnostic assay for the agent of interest.

Metagenomic sequencing of a biological sample can potentially identify all the pathogens involved in a disease condition, but these samples are always associated with a large amount of host genomic sequence data which can reduce the test sensitivity for detection of pathogens in comparatively small numbers compared to host material in the sample. With this type of testing, it can be difficult to determine the significance of all the organisms detected in relation to disease diagnosis in a clinical system [5]. However, clinical metagenomics is a comparatively newer technique in which the clinical sample is subjected to targeted sequencing to reveal all the pathogens of interest in the clinical sample [6,7,8]. In the current study, we developed a targeted next-generation sequencing-based panel for the identification of pathogens involved in feline URD which eases the necessity of performing multiple tests. The assay includes SARS-CoV-2, to allow screening for this less commonly detected pathogen in cats, while also testing for the more common feline respiratory pathogens. The assay also produces genomic information for the pathogens, which gives an added benefit over conventional molecular diagnostic methods by providing confirmation of positive results. Analytical sensitivity and specificity were evaluated for SARS-CoV-2 and the other more common feline URD pathogens. Diagnostic sensitivity and specificity were also analyzed using clinical samples previously tested for these organisms by qPCR assays. The developed panel, employing targeted next-generation sequencing for pathogen identification in feline URD, not only simplifies testing by consolidating multiple pathogens into a single assay but also enhances diagnostic accuracy. Considering ongoing human SARS-CoV-2 cases, the improved diagnostic capabilities of this panel play a crucial role in swiftly and precisely detecting potential infections within feline populations through a comprehensive single test which includes SARS-CoV-2.

## 2. Materials and Methods

### 2.1. Primer Design for Targeted NGS Assay

The primers used in the study were designed in conjunction with the Agriseq Bioinformatics team (Thermofisher Scientific, Austin, TX, USA). They included a wide array of primers targeting multiple genes and regions of genes for approximately 70 different pathogens in pet animals, with approximately 350 different primer targets for around 160 different pathogen genes, including pathogens associated with feline URD. Primers were designed to detect more than one area of the genome for each pathogen to ensure detection, particularly with expected differences in primer function. Additionally, highly conserved areas of each genome were chosen for primer design. For example, for SARS-CoV-2, conserved portions of the envelope (E), nucleocapsid (N), and RNA-dependent RNA polymerase (RdRp) genes were targeted, based on alignment of multiple variants. The targeted regions were approximately 200 nucleotides in length, to be consistent with the requirements for the sequencing technology we used. The primers were distributed into two different pools to minimize primer interactions and to give maximum amplification in the initial step of the Thermofisher Ion Ampliseq^®^ sequencing technology. A FASTA file (Appendix A) containing the genomic sequence of the targeted pathogens and a gene target BED file (Appendix A) was also uploaded into the Torrent suite software (TSS version, 5.16.1, ThermoFisher Scientific, Waltham, MA, USA) for aligning and analyzing the sequenced read results. The designed primer pools are available for purchase from ThermoFisher Scientific (Austin, TX, USA).

### 2.2. Library Preparation and Targeted NGS Assay

An automatic library preparation using Ion Chef™ Instrument (ThermoFisher Scientific) or manual library preparation was performed in-house for loading the semiconductor chip, and sequencing was performed in an Ion GeneStudio™ S5 System (ThermoFisherScientific). The protocol used in this study is documented in detail at (https://dx.doi.org/10.17504/protocols.io.j8nlkk75wl5r/v1 (accessed on 20 December 2023)) hosted in protocols.io [9]. A brief protocol is as follows: A cDNA library was prepared from the nucleic acid extracted from 200 µL of the oronasal swab supernatant. Following the manufacturer’s protocol, 10 µL of cDNA was prepared from each sample using NGS Reverse transcriptase (ThermoFisher Scientific). The manual library preparation was performed using the Ion AmpliSeq™ Library Kit Plus (ThermoFisher Scientific), and the reaction volumes were adjusted based on the manufacturer’s protocol (Ion AmpliSeq™ Library Kit Plus, MAN0017003). An initial PCR amplification was carried out with the synthesized pool of primers (pool 1 and pool 2). About 7.5 µL of sample-specific cDNA and 5 µL of 5X Ion AmpliSeq™ HiFi Mix were mixed and two 5 µL sample aliquots were run in two separate reactions. The PCR cycling condition was standardized for 99 °C initial denaturation, followed by 27 cycles of conditions with a 15 s cyclic denaturation at 99 °C, followed by 60 °C annealing/extension. Later, the two reactions were mixed for each sample and partially fragmented using FUPA reagent provided in the kit at 50 °C for 10 min, 55 °C for 10 min, and 60 °C for 20 min. This was followed by a reaction for the ligation of separate barcodes (Ion Xpress™ Barcode Adapters 1-16/17-32/33-48 Kit, ThermoFisher Scientific) in an incubation temperature of 22 °C for 30 min, 68 °C for 5 min and 72 °C for 5 min for differentiating each sample in the final library. The final library was purified using AMPure XP (Beckman Coulter, Brea, CA, USA) and eluted in 25 µL of low TE buffer. The quantity of the purified barcoded library was colorimetrically estimated using Qubit™ 1X dsDNA High Sensitivity assay kit on a Qubit 4 Fluorometer (ThermoFisher Scientific).

Ion AmpliSeq™ Kit for Chef DL8 and IonCode™ Barcode Adapters 1-32 (ThermoFisher Scientific) was used for automatic library preparation protocol. The 10 µL of the initial cDNA was mixed with 5 µL of nuclease-free water and loaded onto the first column wells of the IonCode plate. The rest of the automatic library preparation was carried out on the IonChef instrument following the manufacturer’s protocol. A 50 µL final volume of an equimolar solution of multiple barcoded libraries (approx. 100 pM) was used to load an Ion 530™ Chip (ThermoFisher Scientific) for both the manual and automatic prepared libraries using an Ion 510™ & Ion 520™ & Ion 530™ Kit—Chef on the Ion Chef instrument following the manufacturer’s protocol. Multiplexed libraries were loaded onto the chip to achieve at least 500,000 reads per sample within the maximum limit of 48 million reads per Ion 530 chip, with ideal loading conditions. The loaded Ion 530 chips were sequenced using Ion S5 sequencing reagents (ThermoFisher Scientific) according to the manufacturer’s protocol. The initial curation of the data was performed in the TSS using SPAdes (v 5.12.0.0) and mapped to the reference FASTA files (Appendix A) and the target region BED file (Appendix A) uploaded in the TSS. The mapped aligned sequencing reads were downloaded as BAM files from the TSS and viewed and analyzed in Geneious Prime software (Boston, MA, USA), v. 2021.1.1 (https://www.geneious.com/, accessed on 20 December 2023) and the results were confirmed using BLAST analysis (Bethesda, MD, USA) (https://blast.ncbi.nlm.nih.gov/Blast.cgi, accessed on 20 December 2023). Samples were considered positive if we were able to detect at least 10 reads with more than one primer set. This was determined through our limit of detection testing.

### 2.3. Feasibility, Analytical Sensitivity, and Analytical Specificity of the Assay for Respiratory Pathogens

The feasibility of the assay for each organism was explicitly evaluated. For SARS-CoV-2, the assay was evaluated by testing the feline matrix (nucleic acids extracted from feline oropharyngeal/nasal swabs and respiratory tissues stored at −80 °C prior to the appearance of SARS-CoV2) spiked with 10-fold serial dilutions of stored nucleic acids extracted from all known variants available in human SARS-CoV-2 samples at the time, including the Omicron variant. The SARS-CoV-2 human samples previously tested positive at the Animal Disease Diagnostic Laboratory (ADDL), Purdue University by qPCR using the TaqPath COVID-19 Combo Kit (ThermoFisher Scientific), and variants were determined by whole genome sequencing performed on an Ion GeneStudio S5 System (ThermoFisher Scientific) using the Ion AmpliSeq SARS-CoV-2 Insight Research Assay (ThermoFisher Scientific) as described below. For other organisms known to cause respiratory disease in cats, we used ATCC strains (Manassas, VA, USA) (Table 1).

Previously extracted nucleic acids from clinical samples known to be positive and negative for FeHV-1, FCV, *Bordetella bronchiseptica*, *Mycoplasma* spp., and *Chlamydia* spp. by qPCR testing performed by the ADDL with validated standard operating procedures were used to evaluate the tNGS assay analytical specificity. This testing was based on the ability of the primers in the tNGS assay to detect the correct organisms. The nucleic acids had been stored at −80 °C prior to testing. The positive samples were also used to determine the relative limit of detection (LOD) for each pathogen as described below. The analytical specificity of the primers in the assay was also evaluated in silico by comparing and confirming the obtained sequencing read results for specific pathogens in nucleotide BLAST (https://blast.ncbi.nlm.nih.gov/Blast.cgi?PROGRAM=blastn&PAGE_TYPE=BlastSearch&LINK_LOC=blasthome, accessed on 20 December 2023).

The relative LOD was determined by comparing results obtained from the targeted NGS testing to qPCR results. Positive nucleic acids from clinical samples were serially diluted 10-fold and tested to estimate detection limits.

### 2.4. Ruggedness and Precision of the Assay

To evaluate ruggedness of the assay, 3 different extraction kits (MagMAX Pathogen RNA/DNA Kit, MagMAX™ CORE Nucleic Acid Purification Kit, using bead-beating, and the MagMAX Viral/Pathogen Nucleic Acid Isolation Kit -all from ThermoFisher Scientific, Waltham MA) in a KingFisher Flex Purification System (ThermoFisher Scientific) were tested. Two different reverse transcriptase kits (NGS Reverse transcriptase and SuperScript VILO cDNA Synthesis Kit—ThermoFisher Scientific), as well as both manual and automated library preparation, as previously described, were tested. PCR targeting was tested at both 27 and 30 cycles. We also checked the stability of the samples before extraction and sequencing by subjecting anterior nasal and oropharyngeal swabs sets that were spiked with various concentrations of SARS-CoV-2, FeHV-1, FCV, *B. bronchiseptica*, *C. felis*, and *M. felis*, placed in molecular transport media (Primestore MTM, Longhorn Vaccines & Diagnostics, Bethesda, MD, USA), at both 4 °C and −20 °C for 5 days prior to testing and compared to the same sample set tested on the day it was contrived in the laboratory.

Precision/reproducibility were evaluated using a sample set containing 4 positive samples and 4 negative samples, measured in 3 separate runs over 3 days, at least 1 day apart, by 2 different operators.

### 2.5. Diagnostic Sensitivity and Diagnostic Specificity of the Assay

To evaluate the diagnostic sensitivity and specificity of the assay for SARS-CoV-2, known positive (n = 8) and negative (n = 45) samples from cats (based on PCR testing) were obtained. The samples were procured from an outside laboratory (5 positive samples and 15 negative samples), cats from known COVID-positive households (n = 18), and cats with respiratory disease in crowded shelter situations (n = 15) and were tested by both a SARS-CoV-2 nucleic acid amplification test (NAAT, TaqPath COVID-19 Combo Kit, ThermoFisher Scientific) and by the tNGS assay.

To evaluate the diagnostic sensitivity and specificity of the other pathogens, we used a set of 31 clinical samples previously tested by qPCR at the ADDL to evaluate the tNGS assay. For the positive samples, Ct values ranged from the teens to mid-30s (Appendix A).

### 2.6. Characterization of Positive cat SARS-CoV-2 Samples

For SARS-CoV-2-positive cats, the virus genome from the cat was compared with that from its owner. Sequencing for positive nasal swab samples from both owners and cats was performed with the targeted NGS panel (Ion AmpliSeq SARS-CoV-2 Insight Research Assay, ThermoFisher Scientific), with automated library preparation and chip loading performed on the Ion Chef and sequencing performed with the Ion GeneStudio S5. Sequence analysis was performed with the Ion Reporter Software (v. 5.12.1). Depth of coverage (SARS-CoV-2 coverageAnalysis plugin version 5.16.0.4), mean quality score (FastQC), and generation of consensus sequences (Ion plugin generateConsensus version 5.16.0.10) to determine clade (Nextclade v. 1.13.0) and lineage (Pangolin COVID-19 Lineage Assigner version 3.1.17 [case 1] or Pango v.4.1.1 PLEARN-v1.12 [case 2]) were performed. The sequences were submitted to GISAID for two of the SARS-CoV-2 positive samples.

## 3. Results

### 3.1. Feasibility, Analytical Sensitivity and Specificity

The current assay positively identified all targeted pathogens, including variants of SARS-CoV-2. Samples for feasibility testing had real-time PCR Ct values in the high 20s to low 30s.

Based on the comparison, the relative limit of detection for FeHV-1, FCV, *Mycoplasma* spp., *C. felis*, and *B. bronchispetica* with the targeted NGS panel was at a level equivalent to a Ct value of approximately 35-37 (Table 2). There was some variation in primer function for primers detecting SARS-CoV-2. The primers for the envelope gene consistently detected fewer reads than the primers for the N and RdRp.

The results for the SARS-CoV-2 testing were comparable to an FDA-approved real-time PCR assay (TaqPath COVID-19 Combo Kit, ThermoFisher Scientific), which had a Ct value cut-off for a positive sample at 37.

All the primer sets for each organism were specific for the intended organism. Also, there was no interference with the sample matrix, i.e., feline DNA was not cross-detected by any of the primer sets used in the assay for these organisms.

### 3.2. Ruggedness and Precision of the Assay

Sequencing of four positive and negative samples was highly reproducible when handled by two different operators run over at least 1 day apart using separate sequencing chips, reverse transcriptases (RT), manual and automatic library preps, and separate sequencing kits. No differences were noted with variation in RT kits, manual vs. automated libraries, and use of 27 vs. 30 cycles for the PCR step for targeting; however, the MagMAX™ CORE Nucleic Acid Purification Kit with silica bead beating had lower extraction efficiency, as there was a loss in sensitivity of detection of the SARS-CoV-2 (approximately 10-fold reduction in reads detected at low virus concentration) (Appendix A). Samples containing various concentrations of the feline URD pathogens were found to be stable (detectable by the tNGS assay across a range of concentrations) at 4 °C and −20 °C when compared to the same sample set tested on the day the samples were contrived.

### 3.3. Diagnostic Sensitivity and Specificity

Out of the 53 samples procured from different sources for SARS-CoV-2 testing, all the positive SARS-CoV-2 samples (n = 8, 5 provided by an outside laboratory and 3 detected during this study from cats in SARS-CoV-2 positive households) and negative samples (n = 45) identified by qPCR were accurately identified in the developed tNGS assay. Thus, the diagnostic sensitivity and specificity were 100% based on this limited number of samples. None of the cats that were positive for SARS-CoV-2 had any additional respiratory pathogens. None of the screened shelter cat samples revealed the SARS-CoV-2 genomic reads, whereas 12/15 samples were positive for at least one respiratory pathogen. Among them, *C. felis* was identified in nine samples, FCV in six samples, *B. bronchiseptica* and FeHV-1 in two samples each. The samples were also positive for multiple respiratory pathogens in seven samples, and among them, *C. felis* was found in combination with FCV in four samples. Similarly, FCV and FeHV-1, *C. felis*, and FeHV-1, as well as *C. felis* and *B. bronchiseptica* were seen in combination in one sample.

For other pathogens, there was some variation in diagnostic sensitivity and specificity (Table 3, Appendix A). All the samples used for this testing were negative for SARS-CoV-2 by the tNGS assay.

### 3.4. Characterization of SARS-CoV-2 Positive Cats

Three SARS-CoV-2-positive cats discovered during the study were sequenced. Cat and owner from household 1 (detected 11/11/2021) had delta variant AY.46.4, and cats and owner from household 2 (detected 6/21/2022) had Omicron variant BA.5.2.1. The sequences were submitted to GISAID (https://gisaid.org/ (accessed on 20 December 2023)—cat (EPI_ISL_11348757) and 54-yr-old female (EPI_ISL_11349083)—case 1 from 11/11/2021. Sequences were deposited in GISAID cat 1 (EPI_ISL_13647600) and 42-year-old female (EPI_ISL_13644954)—case 2 from 6/21/2022. A partial sequence was obtained from the other cat (case 2), also lineage BA.5.2.1; however, the sequence was too poor (insufficient coverage) to further evaluate or deposit in GISAID.

## 4. Discussion

In this study, we formulated a tNGS sequencing panel to identify multiple respiratory diseases in cats comprehensively. After the Omicron variant (B.1.1.529), there was a significant decrease in SARS-CoV-2 cases worldwide. However, in light of the appearance of new variants such as BA.2.86 (initially reported in August 2023) and its sub-lineages, it is imperative to maintain ongoing surveillance for SARS-CoV-2 in all susceptible species. The assay’s ability to detect multiple pathogenic agents in a single test, showing comparable agreement for approximate LOD with qPCR results in the Ct range of 35–37, affirms its potential as a valuable screening tool for respiratory pathogens, both common and uncommon, in felines. The PCR amplification step within the tNGS assay enhances its sensitivity by minimizing the effect of unnecessary host genome data in the final read results. In accordance with previous studies [10,11] the developed tNGS assay has a high diagnostic sensitivity when compared to qPCR for several of the evaluated pathogens, including SARS-CoV-2, FeHV-1, FCV, and *M. felis*. However, for C. felis and *B. bronchiseptica*, the sensitivity was reduced compared to qPCR testing. The sensitivity may be dependent upon several parameters, including primer specificity and PCR cycle numbers during the library preparation [12]. A small sample set was used to evaluate the diagnostic sensitivity and specificity of the assay for the more common feline respiratory pathogens. The small number of samples likely skewed the results, but the discrepancy in the sensitivity between results obtained with qPCR testing versus the tNGS assay was high for *B. bronchiseptica* and C. felis. The qPCR Ct values for the discrepant *Bordetella* samples were 33.21 and 35, suggesting the potential to miss samples with low amounts of organisms with the tNGS assay. The approximate LOD for this organism with the tNGS assay was determined to be 35, so this is likely. Additionally, these samples had been stored for a period at −80 °C, so with freeze–thaw, the number of detectable organisms may have been lower than when originally tested. Interestingly, the tNGS assay was able to detect *B. bronchiseptica* in one sample that was missed by the qPCR testing. There were sequences obtained from two different targeted regions of this organism from this sample (each sequence approximately 200 bp in length), and both had 100% coverage and identity to *B. bronchiseptica* with BLAST analysis, suggesting this was a true positive detection.

The discrepancy with the *C. felis* results was unexpected, based on LOD testing. Samples positive by qPCR but negative by the tNGS assay had Ct values of 27.69, 30.42, 31.33 and 31.97. These Ct values are well below the approximate LOD determined for this organism for the tNGS assay. These particular discrepant samples were tested twice by tNGS and a second time by qPCR to confirm the results. The discrepancy needs further evaluation to determine if the issue is with the qPCR or with the tNGS assay. The qPCR assay is not specific for *C. felis* (detects *Chlamydia* spp.), but the tNGS assay is. Potentially the qPCR assay is detecting a different *Chlamydia* species in these samples. Alternatively, there is some issue with the sensitivity of the tNGS assay that was not apparent with the analytical sensitivity testing. The addition of more primer sets for *Chlamydia* spp. may improve the assay for the detection of this pathogen.

The diagnostic specificity of the assay was 92–100% for all the pathogens tested except for *M. felis.* Based on the sequences obtained for these discrepant samples with the tNGS assay, we believe the tNGS assay is truly more sensitive for the detection of *M. felis* than the qPCR assay used by the ADDL.

Overall, the assay was suitable for screening various respiratory diseases, including different SARS-CoV-2 variants. Coinfection with multiple pathogens was noticed in sheltered cats, where the combination of *C. felis* with FCV was notable. Although not considered a virus associated with respiratory disease, our panel also identified the immunosuppressive feline leukemia virus in 7 out of the 15 screened shelter cat samples.

Without reducing the assay’s sensitivity, we could also replicate identical results when the samples were stored at different storage temperature conditions, like 4 °C and −20 °C. This was expected due to the use of MTM, which is reported to stabilize RNA at room temperature. The assay was also checked for its ruggedness and precision by using different nucleic acid extraction methods, RTs, library preparation methods, and operator variations in multiple sequencing runs. The only variation noted was of low sequencing efficiency when MagMAX™ CORE Nucleic Acid Purification Kit was used. This is attributed to possible RNA degradation due to the bead beating step which applies force to disrupt the cells. The bead-beating step was included to try to improve the DNA extraction efficiency of bacterial samples. Simner and colleagues [13] reported a parallel observation, noting increased viral sequence reads from cerebrospinal fluid where the bead-beating step was excluded in extraction methods. In contrast, bacterial samples exhibited a high number of reads when combined with a bead-beating step. Recent testing with the MagMAX^TM^ CORE kit using swabs for extraction without bead-beating has shown it to be as efficient as the other extraction methods used in the study for extraction of low amounts of SARS-CoV-2. 

In the study, among the positively tested SARS-CoV-2 cat samples, one case was received from a household where the owner tested positive for Coronavirus Disease-19 (COVID-19) by a nucleic acid amplification test under an Emergency Use Authorization (TaqPath COVID-19 combo kit, ThermoFisher Scientific) with very low Ct values of 12.8, 13.6 and 13.18 for N, ORF1ab and S genes, respectively. The household had five cats, whereas only one cat tested positive for SARS-CoV-2 with Cts of 26, 26.7 and 26.9 for N, ORF-1ab, and S genes, respectively. Based on whole genome sequencing, the owner and the cat had the same virus genome, and both viruses were designated in Clade 21J (delta) (Nextclade v. 1.13.0) and as lineage AY.46.4 (Pangolin COVID-19 Lineage Assigner version 3.1.17). Both the sequences had 43 variations (40 SNP and 3 deletions) from the original SARS-CoV-2 isolate, resulting in 31 missense and 5 silent mutations. Due to the higher viral load, the owner had approx. 300 thousand reads, and the cat had only 5 thousand reads of the SARS-CoV-2 genes. Based on the time frame and the quality of the results, and no infection in other cats in the household, we believe that the owner had the first infection and transmitted the disease to the cat, not vice versa. While there are numerous reports of domestic cats contracting SARS-CoV-2 and experimental transmission of SARS-CoV-2 from cat-to-cat [14], it remains unclear whether they can transmit the disease to humans, with evidence suggesting either negligible or undetermined transmission potential [15]. In our observation involving the household with five cats, none of the other cats sharing the living space with the SARS-CoV-2-positive cat showed any signs of infection. In the second case, we could not conclude human-to-cat for both positive cats or cat-to-cat transmission, as the sequences obtained for the second cat were of poor quality and were not used for downstream analysis. Both households had positive cases of human COVID and had histories of close contact, such as cuddling and sleeping in the same bed between the cat and the owner. In conclusion, the developed tNGS assay is a cost-effective method for large-scale screening of respiratory pathogens in comparison to deep sequencing, and it offers the flexibility of multiplex PCR targeting to identify an array of pathogens in a single test with added genomic confirmation.

## Figures and Tables

**Table 1 pathogens-13-00335-t001:** Sample details and strain names of different target respiratory pathogens used as positive controls in targeted NGS assay.

Target Pathogen	Source	Strain/Isolate Detail
FeHV-1	ATCC	FVRm KL 6.19.13
FCV	ATCC	FCV 2280
SARS-CoV-2	qPCR Positive ^1^	Multiple ^2^
*Mycoplasma* spp.	ATCC	23391
*Chlamydia felis*	ATCC	FP Baker; VR-120
*Bordetella bronchiseptica*	ATCC	19395

^1^ TaqPath COVID-19 Combo Kit assay performed at Animal Disease Diagnostic Laboratory, Purdue University; ^2^ 11 SARS-CoV-2 variants including AY.39, AY.3, AY. 103, AY.44, B.1.617.2, BA.1.1, BA.1, BA.2, BA.2.9, BA.2.12, BA.2.12.1 sublineages were used for validation.

**Table 2 pathogens-13-00335-t002:** Relative limit of detection for different target pathogens identified in targeted NGS assay.

Target Pathogen	tNGS Detection	Ct Value (LOD Value)
FeHV-1	Detected	36
FCV	Detected	36
SARS-CoV-2	Detected	37
*Mycoplasma* spp.	Detected	36.5
*Chlamydia felis*	Detected	36.9
*Bordetella bronchiseptica*	Detected	35

**Table 3 pathogens-13-00335-t003:** Diagnostic sensitivity and specificity of the tNGS assay for detection of common feline respiratory pathogens.

Organism	Positive	Negative	Sensitivity	Specificity
FeHV-1	10/10	15/15	100%	100%
FCV	9/10	13/13	90%	100%
*B. bronchiseptica*	5/7	12/13	71.43%	92.31%
*Mycoplasma felis*	12/12	10/12	100%	83.33%
*Chlamydia felis*	3/7	14/14	42.86%	100%

## Data Availability

The protocol for this assay can be found on Protocols.io (https://www.protocols.io/view/targeted-ngs-feline-respiratory-panel-including-sa-j8nlkk75wl5r/v1, (accessed on 20 December 2023)).

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
