# Peer review of "Development of a Targeted NGS Assay for the Detection of Respiratory Pathogens including SARS-CoV-2 in Felines"

_pathogens, 2024, doi:10.3390/pathogens13040335_

Round 1
Reviewer 1 Report
Comments and Suggestions for Authors
In the manuscript entitled “Development of a targeted NGS assay for the detection of respiratory pathogens including SARS-CoV-2 in felines,” the authors provide the framework for a diagnostic assay capable of identifying major microbial causes of feline respiratory disease. While the NGS approach appears to have high diagnostic value, the message in this report appears to be divided between providing a general diagnostic tool to evaluate feline respiratory diseases as well as further work up a few cases in which SARS-CoV-2 has been transmitted from owners to their cats. However, neither focus appears to be fully realized. The reasons outlined below have contributed to my decision to require revisions before acceptance.
Major comments:
1. In general, I believe there needs to be a reassessment of the focus of the manuscript. As written, it feels incomplete and a little confusing. The title alludes to a diagnostic NGS assay that can identify multiple pathogens, but the majority of the discussion is limited to the molecular identity of SARS-CoV-2 of a few cases in domestic cats. This reviewer believes that providing a more complete workup of the respiratory pathogens should be performed without a particular attention to SARS-CoV-2. What is the genome coverage for all pathogens? What is the sequencing depth achieved? Do all primer sets perform equally well? What determines positivity; is it sequence at one locus? What is the cost of the assay compared to the gold standard? In lines 74-77, the authors state that analytical sensitivity and specificity is high. It would be nice to also assess diagnostic sensitivity and specificity in order to highlight the feasibility of this approach in diagnostic settings. Accordingly, the discussion should be expanded.
2. The study is very focused on SARS-CoV-2 variants. Are the primer sets designed focused on conserved regions? What is the likelihood they can pick up emerging variants?
Minor comments:
Line 13 – “variantshas” needs a space
Line 47 – This reference does not appear to directly support the claims that feline leukemia virus and feline parvovirus coupled with co-infections of respiratory pathogens, specifically lead to severe respiratory illness.
Line 112 – there are two spaces between “in two”
Line 134 – I believe this should read “curating”
Table 2 and throughout the manuscript – standardize FeHV-1 and feline rhinotracheitis virus by using one term. This reviewer prefers FeHV-1.
Table 2 – italicize bacteria species names
Line 283 has two spaces between “low sequencing”
Line 286 – there are two periods in a row
Lines 304-306 – this sentence does not make sense.
310 – change “cat-cat” to “cat-to-cat”
Author Response
Reviewer 1
Major comments:
- In general, I believe there needs to be a reassessment of the focus of the manuscript. As written, it feels incomplete and a little confusing. The title alludes to a diagnostic NGS assay that can identify multiple pathogens, but the majority of the discussion is limited to the molecular identity of SARS-CoV-2 of a few cases in domestic cats. This reviewer believes that providing a more complete workup of the respiratory pathogens should be performed without a particular attention to SARS-CoV-2. What is the genome coverage for all pathogens? What is the sequencing depth achieved? Do all primer sets perform equally well? What determines positivity; is it sequence at one locus? What is the cost of the assay compared to the gold standard? In lines 74-77, the authors state that analytical sensitivity and specificity is high. It would be nice to also assess diagnostic sensitivity and specificity in order to highlight the feasibility of this approach in diagnostic settings. Accordingly, the discussion should be expanded.
Thanks for your suggestions. We actually do have the data for the other pathogens and have now included that data in the manuscript. We specifically focused on the SARS-CoV-2 because the RFP was focused on its detection, and it was that grant that funded the study.
Genome coverage- 200 bp portions of the genomes (due to the small read sequencing technology), but we have multiple 200 bp areas targeted for each pathogen. Sequence depth- this varies, based on the amount of organism present in the sample. We called a sample positive if we could detect at least 10 reads by more than one primer set- this was determined through our LOD testing. Not all the primer sets performed equally well, but they all worked. We added all this info to the manuscript. The amount we charge is $200 for the tNGS.
We did provide the data on diagnostic sensitivity and specificity for SARS-CoV-2. We changed the wording to make this clear. We have now also included this data for the other pathogens.
- The study is very focused on SARS-CoV-2 variants. Are the primer sets designed focused on conserved regions? What is the likelihood they can pick up emerging variants? Yes, the primers focus on conserved regions. We added details about this to the manuscript. I did run an in silico analysis with the most prevalent variant right now- JN.1, and the primers should detect it.
Minor comments:
Line 13 – “variantshas” needs a space Corrected
Line 47 – This reference does not appear to directly support the claims that feline leukemia virus and feline parvovirus coupled with co-infections of respiratory pathogens, specifically lead to severe respiratory illness.
Thank you for catching this. We corrected the reference.
Line 112 – there are two spaces between “in two” Corrected
Line 134 – I believe this should read “curating” Corrected
Table 2 and throughout the manuscript – standardize FeHV-1 and feline rhinotracheitis virus by using one term. This reviewer prefers FeHV-1. Corrected
Table 2 – italicize bacteria species names Corrected
Line 283 has two spaces between “low sequencing” Corrected
Line 286 – there are two periods in a row Corrected
Lines 304-306 – this sentence does not make sense. Amended
310 – change “cat-cat” to “cat-to-cat” Corrected
Reviewer 2 Report
Comments and Suggestions for Authors
The authors developed a trageted-amplicon sequencing method to detect several feline respiratory pathogens. This is a very interesting study that was presented well. I only have one comment on improving it.
The authors highlight the sensitivity and the "specificity" of the study throughout the text. While they present data to show sensitivity based on the high Ct values of the samples successfully detected, there are no data presented for the specificity claims. Perhaps, they could test contrived samples where the targets are spiked in at different concentrations relative to the other targets (Covid at LoD vs 100X LoD Mycoplasma, for example). While sequencing methods are less prone to specificity issues than PCR, it would be good to check whether there is any cross reactivity to other pathogens present at high concentrations.
Author Response
Reviewer 2:
The authors developed a trageted-amplicon sequencing method to detect several feline respiratory pathogens. This is a very interesting study that was presented well. I only have one comment on improving it.
Thank you for your kind words.
The authors highlight the sensitivity and the "specificity" of the study throughout the text. While they present data to show sensitivity based on the high Ct values of the samples successfully detected, there are no data presented for the specificity claims. Perhaps, they could test contrived samples where the targets are spiked in at different concentrations relative to the other targets (Covid at LoD vs 100X LoD Mycoplasma, for example). While sequencing methods are less prone to specificity issues than PCR, it would be good to check whether there is any cross reactivity to other pathogens present at high concentrations.
We accidentally left out the specificity results from the results section. Based on in silico analysis, the primers were specific. We then tested their functionality with known pos/neg clinical samples and also with ATCC strains for the bacterial organisms. Each primer set worked and only detected the intended organisms. We added this information to the manuscript.
We did prepare some contrived samples with different amounts of each organism for the evaluation of the extraction kits. See the supplementary results. We particularly used very low amounts of SARS-CoV-2 and FCV because we were most concerned about the ability to detect these RNA viruses among a large amount of bacteria.
Reviewer 3 Report
Comments and Suggestions for Authors
Reference no 2 is cited in the introduction section, but when the reference was cross checked, couldn't find the justification for that reference, in the original article there is no mention of feline parvovirus or so. The reference should be checked and substituted with other..
There is no sequence coverage figure present in the manuscript or in the supporting materials which could be associated with the NGS data. Also, there should be figures of PCR amplifications.
Comments on the Quality of English Language
In the abstract section, second line grammatical errors, should be space between variantshas
Author Response
Reviewer 3:
Reference no 2 is cited in the introduction section, but when the reference was cross checked, couldn't find the justification for that reference, in the original article there is no mention of feline parvovirus or so. The reference should be checked and substituted with other..
Thanks for identifying this issue. We corrected the reference.
There is no sequence coverage figure present in the manuscript or in the supporting materials which could be associated with the NGS data. Also, there should be figures of PCR amplifications.
Coverage (depth) can vary, depending on the number of organisms in the sample. We consider a sample positive if we can detect at least 10 reads with more than one primer set (we have multiple primer sets for each pathogen in the design). As far as percentage of the organism we are sequencing, that is small pieces- approximately 200 bp, but we are targeting multiple 200 bp regions for each pathogen. We added these details to the manuscript. We are not trying to sequence the entire genome of each pathogen, we are only trying to detect the organisms and confirm the detections.
We do not believe adding figures showing real-time PCR curves would add anything to the manuscript. The real-time assays are not the focus of the paper. If you mean showing PCR amplification (targeting) results prior to the sequencing, we do not run gels after the multiplex PCR prior to library preparation, so we do not have this data.
Round 2
Reviewer 1 Report
Comments and Suggestions for Authors
I appreciate the time and effort the authors have put in to address my previous comments. There are a few considerations that I have suggested below and with these small changes, I believe that this manuscript is ready for publication.
Lines 52-54 – the sentence “the major pathogens causing feline URD remain FeHV-1, …” is slightly awkward. I suggest editing for readability.
Line 59 – “Most respiratory disease are manifested…” is also slightly awkward. Perhaps change “are manifested” to “manifest.”
Line 99 – you added a section stating “primers were designed to detect more than one area of the genome for each pathogen.” I would still like to have an estimate of genome coverage for the stated pathogens, perhaps listed in a (supplemental) table. This may provide additional context for the sensitivity of Bordatella and Chlamydia.
Lines 146-148 – This sentence is a bit strange. It may make more sense to report it as “X number of libraries were multiplexed on an Ion 530 chip to achieve approximately 500,000 reads per sample.” Or something to that effect.
Line 148 – I’m unsure what you mean by “positively” loaded.
Line 183 – What is meant by “a small panel of nucleic acids?” Would “previously extracted nucleic acids from clinical samples” be correct?
Line 229 – This sentence is difficult to understand as written.
Line 253 – To what level do the primers perform differently? Specifics are not provided.
Line 347 – Please rewrite to remove the double-negative “not unlikely”.
Line 351 – The characterization of “100% coverage and identity” is a little misleading. I am assuming that the coverage is based on the entire amplicon of the individual primer sets, but it alludes to complete genome coverage, which I believe is unlikely. Please clarify. Providing a table of percent genome coverage will help with this ambiguity.
Comments on the Quality of English Language
While the quality of English is fine, it could be improved. I have made a small number of suggestions to improve areas where readability, but I suggest careful reading and self-editing.
Author Response
I appreciate the time and effort the authors have put in to address my previous comments. There are a few considerations that I have suggested below and with these small changes, I believe that this manuscript is ready for publication.
Thanks for taking time to have a thorough review of the manuscript. Your previous suggestions were extremely helpful for us in improving the quality of our research manuscript.
Lines 52-54 – the sentence “the major pathogens causing feline URD remain FeHV-1, …” is slightly awkward. I suggest editing for readability.
Thanks for your suggestion and for better readability, the sentence has been changed.
Line 59 – “Most respiratory disease are manifested…” is also slightly awkward. Perhaps change “are manifested” to “manifest.”
Thanks for the suggestion. We have changed it according to your suggestion.
Line 99 – you added a section stating “primers were designed to detect more than one area of the genome for each pathogen.” I would still like to have an estimate of genome coverage for the stated pathogens, perhaps listed in a (supplemental) table. This may provide additional context for the sensitivity of Bordatella and Chlamydia.
Thanks for your query and we value it. The primers in the panel cover specific regions in various pathogens, which are then amplified as short amplicons of approximately 200 bp to align with the short-read sequencing technology of the Ion Torrent sequencing system. The positions of the primers in the gene targets and the pools into which the primers are distributed are provided in the BED file in supplementary data 2.
Lines 146-148 – This sentence is a bit strange. It may make more sense to report it as “X number of libraries were multiplexed on an Ion 530 chip to achieve approximately 500,000 reads per sample.” Or something to that effect.
Thanks for your suggestion. In an ideal experiment, 48 million reads is the maximum we can expect in one sequencing run and 500,000 reads per sample means we can load up to 96 different sample libraries in each chip. However, several different factors, including chip loading percentage and Ion bead- amplicon clonality, can happen during the Ion 530 semiconductor chip loading process. So, each step of the tNGS validation process was conducted as different sequencing runs and variable number of libraries were loaded according to steps we performed. The goal was to get at least 500,000 reads for each sample after multiple trial experiments. The numbers of reads per sample is what is important.
Line 148 – I’m unsure what you mean by “positively” loaded.
This has to do with variables that are introduced during the chip loading process (eg., percentage of chip loading and clonality percentages- based on bead loading). However, we removed the term to reduce any confusion.
Line 183 – What is meant by “a small panel of nucleic acids?” Would “previously extracted nucleic acids from clinical samples” be correct?
Thanks for the suggestion and we have corrected it.
Line 229 – This sentence is difficult to understand as written.
We removed the sentence.
Line 253 – To what level do the primers perform differently? Specifics are not provided.
I am not sure I understand your question. As stated, there were fewer reads (fewer numbers of sequences obtained) with the primers for the envelope compared to primers for the other portions of the genome. The numbers of sequences obtained can vary from run to run- this is not a quantitative method. This does suggest reduced sensitivity of detection for the E primer set.
Line 347 – Please rewrite to remove the double-negative “not unlikely”. Done
Line 351 – The characterization of “100% coverage and identity” is a little misleading. I am assuming that the coverage is based on the entire amplicon of the individual primer sets, but it alludes to complete genome coverage, which I believe is unlikely. Please clarify. Providing a table of percent genome coverage will help with this ambiguity.
Thanks for your valuable suggestion. The 100% coverage and identity that was discussed in line 351 was for the two target regions sequenced for Bordetella bronchiseptica in the BLAST analysis and not based on the whole genome coverage. The size of each sequence was approximately 200 bp. This information was added.
The goal of this method is not to amplify whole genomes. We do not need to amplify whole genomes to determine an organism is present. We are just detecting multiple 200 bp portions. This product size is consistent with this sequencer and these sequences are long enough to provide specificity. The specifics related to the assay can be found in the fasta and bed files, which are provided as supplementary material.
Reviewer 3 Report
Comments and Suggestions for Authors
no comments!
Author Response
Thanks for the time you spent reviewing our manuscript!